# Cell Therapy for Retinal Degenerative Diseases: Progress and Prospects

**DOI:** 10.3390/pharmaceutics16101299

**Published:** 2024-10-05

**Authors:** Kevin Y. Wu, Jaskarn K. Dhaliwal, Akash Sasitharan, Ananda Kalevar

**Affiliations:** 1Department of Surgery, Division of Ophthalmology, University of Sherbrooke, Sherbrooke, QC J1G 2E8, Canada; 2Faculty of Health Sciences, Department of Medicine, Queen’s University, Kingston, ON K7L 3N6, Canada; 3Faculty of Medicine and Health Sciences, Department of Medicine, McGill University, Montreal, QC H3A 0GA, Canada

**Keywords:** stem cells, mesenchymal stromal cells, cell therapy, gene therapy, retinal degenerative diseases, age-related macular degeneration, retinitis pigmentosa, embryonic stem cells, induced pluripotent stem cells, retinal progenitor cells, photoreceptor replacement, neuroprotection, paracrine effects, clinical trials

## Abstract

**Background/Objectives:** Age-related macular degeneration (AMD) and retinitis pigmentosa (RP) are leading causes of vision loss, with AMD affecting older populations and RP being a rarer, genetically inherited condition. Both diseases result in progressive retinal degeneration, for which current treatments remain inadequate in advanced stages. This review aims to provide an overview of the retina’s anatomy and physiology, elucidate the pathophysiology of AMD and RP, and evaluate emerging cell-based therapies for these conditions. **Methods:** A comprehensive review of the literature was conducted, focusing on cell therapy approaches, including embryonic stem cells (ESCs), induced pluripotent stem cells (iPSCs), mesenchymal stem cells (MSCs), and retinal progenitor cells. Preclinical and clinical studies were analyzed to assess therapeutic potential, with attention to mechanisms such as cell replacement, neuroprotection, and paracrine effects. Relevant challenges, including ethical concerns and clinical translation, were also explored. **Results:** Cell-based therapies demonstrate potential for restoring retinal function and slowing disease progression through mechanisms like neuroprotection and cell replacement. Preclinical trials show promising outcomes, but clinical studies face significant hurdles, including challenges in cell delivery and long-term efficacy. Combination therapies integrating gene editing and biomaterials offer potential future advancements. **Conclusions:** While cell-based therapies for AMD and RP have made significant progress, substantial barriers to clinical application remain. Further research is essential to overcome these obstacles, improve delivery methods, and ensure the safe and effective translation of these therapies into clinical practice.

## 1. Introduction

Retinal degenerative diseases, including age-related macular degeneration (AMD) and retinitis pigmentosa (RP), are significant contributors to vision loss, affecting millions of people worldwide. AMD is particularly prevalent among older populations, while RP, though rarer, is a genetically inherited condition that progressively impairs vision. Despite advancements in ophthalmology and ocular pharmacology, effective treatments for these conditions remain limited, especially in the later stages of the disease.

Cell-based therapies have emerged as one of the potential solutions to address the complex challenges of retinal degeneration, offering the potential to replace or repair damaged retinal tissues, thereby restoring vision or at least halting further degeneration. This review provides a comprehensive examination of the anatomy and physiology of the retina, the pathophysiology of AMD, RP, glaucoma, and Stargardt disease, and the latest developments in cell therapy strategies, including the use of embryonic stem cells (ESCs), induced pluripotent stem cells (iPSCs), mesenchymal stem cells (MSCs), and progenitor cells.

A key focus of this review is on the most recent preclinical and clinical studies published within the last few years, reflecting the rapid advancements in the field. We explore the underlying mechanisms of action, such as cell replacement, neuroprotection, and paracrine effects, and discuss the challenges of translating these therapies from the laboratory to the bedside. We discuss future directions, including innovations in cell delivery techniques, combination therapies, and the ethical considerations surrounding stem cell use in retinal treatments. This review article aims to provide a clear understanding of the current state and future potential of cell therapy for retinal degenerative diseases.

## 2. Anatomy and Physiology of the Retina

### 2.1. Structure of the Retina and Choroid

The retina is an intricate structure composed of 10 neuronal layers and 6 different cell lines (Figure 1). Layers are connected to one another via synapses and each cell line plays a role in detecting variations and movements of light [1]. These cells include photoreceptor cells, horizontal cells, bipolar cells, amacrine cells, and retinal ganglion cells.

Photoreceptor cells include rods and cones. Rod cells comprise about 95% of photoreceptors in the retina and register low-light levels, helping to create scotopic vision [1]. Rods are concentrated in the periphery of the retina, with cone cells being concentrated in the retinal center, in the macula. Cone cells comprise about 5% of the retinal photoreceptors. Cone cells aid in processing color vision at various light levels and allow for greater spatial acuity, providing information for fine details, movement, and colors.

Horizontal cells are located between bipolar cells and photoreceptors. They provide inhibitory feedback to bipolar cells, rods, and cones and help the eyes adjust to low light and bright light [1]. Bipolar cells receive glutamatergic input from rods and cones and GABAergic inhibitory input from horizontal cells. They project their axons to retinal ganglion cells, providing glutamatergic inputs. Bipolar cells are present in the inner plexiform layer of the retina and link the outer and inner layers of the retina. Amacrine cells inhibit bipolar cells before the synapse at the inner plexiform layer. Retinal ganglion cells are photosensitive and assist with circadian rhythm, melatonin release, and regulation of pupil size.

Various retinal cells can be damaged in different ocular and retinal diseases. For example, retinitis pigmentosa (RP) primarily affects the rod cells, leading to their degeneration; glaucoma results in the destruction of retinal ganglion cells due to optic nerve damage; and in age-related macular degeneration (AMD), the compromised function of retinal pigment epithelial (RPE) cells leads to photoreceptor loss.

### 2.2. Retinal Blood Supply

The retina has a dual blood supply as it has the highest rate of oxygen consumption in the body. The dual blood supply is received by the choroid and branches of the ophthalmic artery, which arises from the internal carotid artery, giving rise to the central retinal artery and posterior ciliary arteries [1]. The central retinal artery provides blood to the inner retina. The posterior ciliary artery divides into short and long posterior ciliary arteries and provides blood flow to the outer retina. Retinal blood flow is typically low and influenced by local factors such as nitric oxide, prostaglandins, endothelin, and arterial carbon dioxide tension [2]. On the other hand, the choroid has high blood flow and low oxygen content. The choroid forms the posterior part of the uveal tract and receives blood supply from the long and short posterior ciliary arteries, nourishing the outer layers of the retina.

The blood–retina barrier is composed of the inner and outer blood–retinal barriers. The outer blood–retina barrier regulates transportation across the choriocapillaris and the retina, while the inner blood–retina barrier regulates transportation across the retinal capillaries [3]. The outer-blood retina barrier is formed at the RPE [3]. The inner is formed of very tight junctions and consists of Muller cells, which help support barrier function in the retina. An important fact of the neural retina is that it has immune privilege, proving it a suitable site for cell transplantations [4].

## 3. Retinal Degenerative Diseases

Millions of people worldwide are impacted by retinal degenerative diseases, which are a leading cause of vision loss and blindness [5]. This section provides an overview of prominent retinal degenerative disorders: age-related macular degeneration (AMD), retinitis pigmentosa (RP), glaucoma, and Stargardt disease (SD).

### 3.1. Age-Related Macular Degeneration (AMD)

Age-related macular degeneration is a leading cause of blindness globally, especially in the West [5]. It is associated with the degeneration of the macula, a region of the retina important for central vision and home to a large number of cone photoreceptors [6]. The macula also includes the fovea, which is the center of the macula. There are two types of AMD: dry AMD and wet AMD.

More than 90% of AMD patients experience dry AMD, sometimes referred to as nonexudative or non-neovascular AMD [7]. Although it usually advances gradually over decades, it can eventually lead to wet AMD. The thickening of the Bruch membrane resulting from the build-up of drusen (lipid and protein deposits) between the Bruch membrane and the retinal pigment epithelium (RPE) is a characteristic of dry AMD [8]. Retinal degeneration and atrophy are the results of this accumulation [7].

Wet AMD, or exudative/neovascular AMD, accounts for approximately 10–15% of AMD cases but is responsible for the majority of severe vision loss associated with the condition. [7]. Unlike the dry form, wet AMD progresses rapidly, often leading to significant vision loss within weeks to months. The hallmark of wet AMD is the development of choroidal neovascularization (CNV), where new, abnormal blood vessels grow from the choroid through defects in the Bruch’s membrane into the subretinal space. These vessels are prone to leakage of blood, lipids, and fluids, which can cause retinal pigment epithelium (RPE) detachment and photoreceptor damage. The role of the Vascular Endothelial Growth Factor (VEGF) is crucial in this process, as it promotes the growth and permeability of these abnormal blood vessels. The leakage from these fragile vessels leads to the accumulation of fluid and blood, resulting in rapid vision deterioration if left untreated [9].

The incidence of AMD increases after the age of 50. Risk factors for AMD include aging, blue-colored eyes, Caucasian ancestry, family history, sun exposure, smoking, alcohol intake, high blood pressure, obesity, and diabetes [10]. Clinical signs include increasing loss of central vision, trouble focusing on tasks, decreased night vision, trouble adjusting to light, fluctuating vision, and metamorphopsia [11]. Four groups can be used to categorize AMD severity (Figure 2): Group I consists of five to fifteen small lesions; early-stage AMD is characterized by more than fifteen small lesions or fewer than twenty medium-sized lesions; the intermediate stage is characterized by at least one large lesion or more than twenty medium-sized lesions or non-central geographic atrophy (GA); and the advanced stage is characterized by central geographic atrophy or wet AMD [12].

The available preventive treatment options for dry age-related macular degeneration include dietary supplements containing vitamin E, vitamin A, zinc, cupric oxide, lutein, zeaxanthin, and omega-3 fatty acids, as well as lifestyle changes like alcohol and smoking cessation [12]. Anti-VEGF therapy, which involves monthly injections of ranibizumab, bevacizumab, or aflibercept, is used to treat wet AMD [10]. Anti-VEGF therapy improves visual acuity or, at the least, it stabilizes the neo-vascularization response in a larger number of individuals [13].

### 3.2. Retinitis Pigmentosa (RP) (Disease)

Retinitis pigmentosa (RP) is a group of genetic disorders characterized by the degeneration of photoreceptors in the retina, primarily affecting rods more than cones [14]. It is the most prevalent retinal illness that is inherited, and it can be X-linked, autosomal recessive, or autosomal dominant. In total, 1 in 5000 persons have RP, and 1 in 100 people are carriers of the disease.

Mutations cause retinal photoreceptors to undergo apoptosis, which in turn causes neighboring cells to undergo secondary apoptosis [14]. As a result, melanin deposits into perivascular regions and RPE cells separate, resulting in the formation of distinctive pigmented deposits. In spite of the term, apoptosis is the main step; inflammation is negligible. Clinical signs include flashes of light (photopsia), reduced peripheral vision leading to central vision loss, and decreased night vision (nyctalopia), which can proceed to complete night blindness.

Current therapeutic modalities for RP include supplementation with Vitamin A and E, which may slow progression in some cases [14]. The prognosis is generally poor for X-linked RP, while autosomal dominant RP has a better prognosis.

Although we have yet to develop a treatment that can restore vision for those suffering from end-stage geographic atrophy due to severe dry AMD or end-stage RP, ongoing research into cell therapy offers hope and will be discussed in detail in the next section.

### 3.3. Glaucoma

Glaucoma is a collection of eye conditions that, if untreated, frequently cause irreversible vision loss due to gradual damage to the optic nerve [15]. Elevated intraocular pressure (IOP) is the most prevalent cause; however, normal-tension glaucoma, in which optic nerve damage develops despite normal IOP levels, can also happen [16]. The trabecular meshwork’s resistance to the aqueous humor’s outflow usually results in a rise in intraocular pressure [17]. Retinal ganglion cell (RGC) axons, which carry visual information from the retina to the brain, are harmed by this mechanical stress on the optic nerve head caused by the elevated pressure [18]. These axons’ function is disrupted by compression, gradually impairing vision [18]. Damage to the optic nerve mostly affects the optic nerve head, resulting in a distinctive pattern of loss of peripheral vision that may eventually lead to tunnel vision or total blindness [15]. RGC degeneration is believed to be facilitated by oxidative stress, neuroinflammatory processes, and mechanical injury [19]. It is well recognized that inflammatory mediators and free radicals aggravate the damage and hasten the course of glaucoma [19].

In order to halt the advancement of optic nerve damage, the main goal of current therapeutic therapies for glaucoma is to reduce IOP [20]. Medications such as carbonic anhydrase inhibitors, beta-blockers, and prostaglandin analogs, which either increase aqueous humor outflow or decrease fluid production, are commonly used as therapies [20]. A different drainage channel for the aqueous humor can also be created by surgical procedures like trabeculectomy or laser therapy [21]. Nevertheless, the damage already done to the optic nerve cannot be undone by these treatments; they can only control the illness. In addition, patients may find it difficult to follow long-term prescription regimens, and complications from surgery are a possibility. Therefore, the creation of cell-based treatments appears promising for those suffering from glaucoma. The goal of stem cell treatments is to repair or preserve injured retinal ganglion cells while regaining the function of the optic nerve. Although these treatments are still in the preliminary stages, they present a viable future option that might not only cure optic nerve damage and restore lost cells, but also halt the progression of the disease [22].

### 3.4. Stargardt Disease (SD)

The most prevalent type of inherited macular degeneration is called Stargardt disease, and it is mainly brought on by mutations in the ABCA4 gene [23]. The protein that this gene produces is in charge of removing harmful byproducts from photoreceptor cells during the visual cycle [24]. These byproducts, especially lipofuscin, accumulate in the retinal pigment epithelium (RPE) as a result of mutations in ABCA4. Both the RPE cells and the photoreceptor cells that depend on the RPE for vital metabolic support are severely damaged by the accumulation of lipofuscin [24]. Cone photoreceptors in the macula, which are in charge of central vision, gradually deteriorate as a result of lipofuscin buildup. The end outcome of this process is a progressive loss of color vision, light sensitivity, and visual acuity [23]. Lipofuscin’s harmful actions exacerbate the degeneration by hindering the RPE’s capacity to preserve the health of the photoreceptor cells [24]. As a result, peripheral vision usually remains intact, while central vision gradually disappears [23]. Due to the degenerative nature of Stargardt illness, many people have severe vision loss early in life, especially in youth [23].

Currently, Stargardt disease cannot be stopped from progressing further or reversed using an approved medication [23]. Supportive therapies may assist with managing symptoms but do not address the underlying pathophysiology. Examples of these therapies include the use of low-vision devices and light protection [23]. Clinical trials are still in progress for gene therapy to fix abnormalities in the ABCA4 gene [25]. The shortcomings of existing treatments highlight the possibility of cell-based therapeutics as a future modality. Damaged retinal tissue may regain its ability to function through cell replacement therapies, such as the transplanting of healthy RPE or photoreceptor cells made from stem cells. While more investigation is needed to address issues with cell survival, integration, and immune response, cell-based therapies offer patients with Stargardt illness a promising path toward retinal cell regeneration and vision preservation.

## 4. Types of Cell Therapies

With the potential to replace or repair damaged retinal cells, cell therapy holds great promise for the treatment of retinal degenerative illnesses [26]. The several kinds of cell therapies (Figure 3) that are presently being researched and developed are examined in this section.

### 4.1. Retinitis Pigmentosa (RP)

#### 4.1.1. Embryonic Stem Cells

Embryonic stem cells (ESCs) are pluripotent cells derived from early-stage embryos. They have the capacity to differentiate into any cell type, including specialized retinal cells such as photoreceptors, retinal pigment epithelial (RPE) cells, and ganglion cells. This versatility makes ESCs a vital tool in regenerative medicine, notably for treating complex retinal degenerative diseases like age-related macular degeneration (AMD) and retinitis pigmentosa (RP). Although ESCs are very adaptable, transplanting them into patients carries a risk of immunological rejection and ethical issues [27]. Since ESCs are usually not produced from the patient’s own cells, the transplanted cells run the risk of being recognized as foreign by the immune system, which could lead to an immunological reaction against them [27]. This may result in the transplanted cells being rejected, which would lessen the therapy’s efficacy and possibly inflict further retinal tissue damage. Immunosuppressive medication carries its own risks and problems, including greater susceptibility to infections and other immune-related conditions, and may be necessary for patients in order to reduce this risk [27].

#### 4.1.2. Induced Pluripotent Stem Cells

Induced pluripotent stem cells (iPSCs) are generated by reprogramming adult somatic cells to a pluripotent state, similar to ESCs [28]. Since iPSCs can be generated from the patient’s own cells, the risk of immunological rejection is reduced, and they avoid many of the ethical concerns that are related to ESCs [29]. Immunosuppressive medications, which are frequently necessary with ESC-based treatments, are not as likely to be needed when these autologous iPSCs are developed into retinal cells and transplanted back into the patient because the immune system is less likely to perceive them as alien [29]. Retinal cells can be created in vitro using iPSCs, and these cells could potentially be grafted into the injured retina to replace any missing or malfunctioning ones [29]. Furthermore, by using iPSCs to simulate retinal diseases in the lab, researchers can better understand the underlying mechanisms of these disorders and create novel treatment approaches.

#### 4.1.3. Mesenchymal Stem Cells

Mesenchymal stem cells (MSCs) are known for their immunomodulatory properties and ability to differentiate into various cell types [30]. Through immune response modulation and trophic support of injured retinal cells, MSCs have the ability to support retinal regeneration and repair [30]. The ability of MSC-derived factors, cells, and modified MSCs to repair injured retinal tissue has been the subject of numerous investigations. Preclinical models of retinal degeneration demonstrated that human dental pulp-derived MSCs (DP-MSCs) improved retinal function in a rat model of retinal degeneration through intravitreal transplantation, while rat bone-marrow-derived MSCs (BM-MSCs) restored the thickness of the outer nuclear layer (ONL) by increasing autophagy [31]. Injections of umbilical cord-derived MSCs (UC-MSCs) and their exosomes improved visual functions and decreased inflammation and retinal damage in a mouse model of intravitreal retinal injury [31].

### 4.2. Progenitor Cell-Based Therapies

Progenitor cell-based therapies represent a promising avenue for treating retinal degenerative diseases by harnessing the regenerative potential of cells that are more differentiated than stem cells but still have the capacity to develop into specific types of retinal cells. This method focuses on replacing or repairing damaged retinal tissue with neural progenitor cells (NPCs) and retinal progenitor cells (RPCs).

#### 4.2.1. Retinal Progenitor Cells

Specialized cells called retinal progenitor cells are derived from the growing retina and have the ability to differentiate into several types of retinal cells, such as photoreceptors and retinal ganglion cells [32]. RPCs have been found to be important in regenerative therapy for retinal illnesses and to play a critical function during retinal development. RPCs are a strong contender for therapeutic intervention because of their capacity to develop into vital retinal cells and blend in with the current retinal architecture [32]. RPCs have demonstrated the ability to halt the course of disease, restore vision, and replace missing or damaged retinal cells when implanted into the retina [32]. RPC transplantation is undergoing clinical trials, and preliminary findings suggest that these cells can proliferate, migrate, and differentiate within the host retina [32].

#### 4.2.2. Neural Progenitor Cells

Neural progenitor cells (NPCs) are multipotent cells that can differentiate into various neural cell types, including neurons, astrocytes, and oligodendrocytes [33]. NPCs are particularly interesting in the context of retinal degenerative illnesses because they can replace retinal neurons while simultaneously performing vital supporting roles that keep the retina healthy [34]. NPCs have been shown to be able to adapt to the retinal environment after transplantation, and they can come from different parts of the central nervous system, such as the brain and spinal cord. Through differentiation into retinal cells and the formation of synaptic connections with pre-existing retinal neurons, NPCs have been demonstrated in preclinical investigations to restore some function of the visual system [33]. To further increase their therapeutic potential, NPCs can release neurotrophic substances that support the survival and functionality of the remaining retinal cells [33].

RPCs and NPCs in particular are progenitor cell-based therapies that provide a focused method for retinal restoration. In addition to replacing lost cells, these therapies work to foster an environment that supports the retina’s long-term survival and performance.

### 4.3. Gene-Edited Cell Therapies

Gene-edited cell therapies are emerging as a revolutionary approach to treating retinal degenerative diseases by directly targeting and correcting genetic mutations responsible for these conditions. A variety of techniques for retinal gene therapy may be employed, contingent upon the kind of mutation: gene replacement or augmentation, editing or silencing the defective gene, or introducing a gene that alters the downstream or upstream pathways from the damaged gene to improve cellular function [35]. Retinal gene therapies employ several vectors and delivery systems. A plethora of gene-editing techniques have been developed, including zinc finger nucleases (ZFNs), transcription activator-like effector nucleases (TALENs), homing endonucleases or meganucleases, and CRISPR/Cas9 [35]. CRISPR/Cas9 is one of the widely used gene-editing tools in biomedical research and there are several gene therapies using this technology in clinical trials [35].

Gene therapy has long been thought to be a great fit for the retina. Advantages include a restricted, immune-privileged area protected by the blood–retina barrier [35]. Since the retina is small and does not proliferate cellularly in adults, retinal disorders can be treated with low dosages of the vector [35]. Products for ocular gene therapy might be administered by clinical protocols or established surgical methods. Nevertheless, despite these advantages, gene therapy faces a major challenge due to the enormous genetic complexity of inherited retinal illnesses, which can involve hundreds of mutations across numerous distinct genes [35]. Due to this variability, developing therapies that are one size fits all is difficult. Furthermore, a precise genetic diagnosis is necessary to pinpoint the precise mutations causing the illness, yet many patients are still without a conclusive genetic diagnosis. Without this vital information, successful gene therapy customization is difficult, which reduces the potential benefit of the treatments.

## 5. Mechanisms of Action

Cell-based therapies for retinal degenerative diseases rely on several key mechanisms of action to restore vision and prevent further damage to the retina. These mechanisms include cell replacement, neuroprotection, and paracrine effects, each contributing to the overall therapeutic potential of these advanced treatments.

### 5.1. Cell Replacement 

As previously discussed, AMD and RP are characterized by the degeneration of photoreceptors and RPE cells, respectively. Consequently, the primary objective of cell-based therapy is to restore retinal function by replacing these damaged or lost cells. This approach aims to replenish the retina with healthy, functional cells that can re-establish the intricate processes of light detection and signal transmission, ultimately preserving or even restoring vision. To achieve this, stem cells or progenitor cells must be differentiated into specific retinal cells.

There exists an optimized protocol to differentiate human-induced pluripotent stem cells into retinal pigment epithelium (RPE) cells [36]. The RPE cells generated following this protocol are mature and have similar cellular and molecular properties to primary RPE cells. Furthermore, the protocol includes an enrichment step enabling large-scale GMP manufacturing, which highlights the potential for cell replacement therapies in treating AMD [36].

Human embryonic stem cells (hESC) can also serve as a stem cell source for RPE cells; however, allogeneic hESC-RPE cells can trigger immune rejection, despite the eye being considered an immune-privileged site. Petrus-Reurer et al. established that hESC-RPEs lacking HLA-I and -II, which have reduced T-cell response in vitro, do not increase NK cell cytotoxic activity, and xeno-transplanted show reduced rejection in a large-eyed animal model [37].

The potential of photoreceptor cells derived from different human iPSC sources, including blood, fibroblasts, and keratinocytes, has been investigated in numerous research studies. However, due to the challenges in developing reliable, effective, and stable techniques for the production and purification of photoreceptor cells, there is no report of the transplantation of iPSC-derived photoreceptor cells in humans for the purpose of vision restoration [38].

### 5.2. Neuroprotection and Paracrine Effects

MSCs derived from bone marrow, umbilical cords, adipose tissues, and human neural progenitor cells take on a trophic role in stem cell therapy. MSCs are known to rescue degenerating photoreceptors via paracrine factors released by the cells. These cells suppress the immune response and inflammation by releasing immunomodulatory proteins such as Th2-related cytokines, insulin-like growth factor-1, and class II major histocompatibility complex antigens. Since PRs undergo mutations that produce RP, delaying the progression of vision loss can be achieved through cell preservation techniques. However, only when sufficient PRs are present in the early stages of the disease are the preservation techniques effective. RPE produced from MSCs can also be utilized as supporting cells to give PR that are still alive trophic support [39].

## 6. Cell Therapy for Retinal Degenerative Diseases

Cell therapy for retinal degenerative diseases has been tested in multiple preclinical and clinical trials over the years. These studies have been conducted in various animal models and looked at diseases from AMD to RP, glaucoma, and retinal degeneration in general. Studies have investigated cell therapy using embryonic stem cells (ESC), induced pluripotent stem cells (iPSC), RPE stem cells, bone marrow, mesenchymal cells, and more (Figure 4). Here, we will provide a review of the preclinical (Table 1) and clinical (Table 2) studies using cell therapy for retinal diseases conducted to date.

### 6.1. Preclinical Studies

#### 6.1.1. Preclinical Studies Using ESCs

Some of the initial studies showing promise for the use of cell therapy in retinal disorders showed rescue and prevention of photoreceptor degeneration by transplanting retinal pigment epithelium (RPE) in Royal College of Surgeon (RCS) rats [40,41,75]. Limitations of these methods included the need for healthy RPE. Later, Schraermeyer and colleagues [42] transplanted ESC and found it to delay photoreceptor degeneration in RCS rats, making ESC a potential source for cell transplantation in retinal diseases.

While there was promise for the use of mouse ESCs, there had been no reports showing the use of primate ESC until Haruta and colleagues [44] investigated the generation of epithelial cells from primate embryonic cells. Embryonic cells were obtained from cynomolgus monkeys, differentiated into embryonic stem-cell-derived pigment epithelial cells (ESPE), and transplanted into the subretinal space of RCS rats, resulting in photoreceptor death and vision loss [44]. After transplantation, the RCS rats were observed to have recovery and retinal function, providing evidence for the use of ESPEs for cell-replacement therapy for retinal degenerative diseases [44]. An advantage of using ESCs for degenerative diseases is that they have the capacity to indefinitely differentiate into any cell type. 

Further studies investigated the use of human ESCs (hESCs) for retinal diseases and found hESC-derived RPE to exhibit morphology, marker expression, and function of authentic RPE, rescuing retinal function in animal models of retinal degeneration [46,47,49]. In RCS rats, an improvement in visual performance was observed compared to untreated controls, after hESC-derived RPE transplantation [46]. Idelson and colleagues [49] confirmed that retinal rescue was not a nonspecific effect by also transplanting human fibroblasts into the subretinal space. Transplantation of fibroblasts did not result in protection of the photoreceptor layer and delay in degradation [49]. Similarly, in *Crx*^−/−^ mice (a model of Leber’s Congenital Amaurosis), hESC-derived retinal cells differentiated into functional rod and cone photoreceptors and restored light responses in the animals [50].

Additional studies found ESC to also possess the potential to differentiate into cells similar to retinal ganglion cells (RGC). In this study, neural progenitors (NP) were first derived from FGF2-induced ESC cells, which then differentiated into RGC-like cells, expressing RGC regulators and markers, such as Ath5, Brn3b, RPF-1, Thy-1, and Islet-1, in vitro [51]. The ESC-NP cells were then exposed to FGF2, which, upon transplantation, integrated and differentiated into RGCs in vivo [51]. This research provided a method for differentiating ESC into RGC and showed efficacy in vivo. Other research has successfully induced rat ESCs into RPEs and photoreceptors, restoring visual function in RCS rats after retinal transplantation [61].

Comparison of subretinal transplantation of mouse ESC-derived rod photoreceptors in mild retinal degeneration and severe retinal degeneration mice showed differences between the two models. The mice ESCs integrated into the mild retinal degeneration models and acquired mature morphology expressing photoreceptor markers, whereas, in severe retinal degeneration models, the transplanted cells survived but did not have mature morphologic features [58]. This may have been due to severely degenerated retinas creating a hostile environment and activated microglia resulting in immune responses and rejection [58]. This study highlights a primary concern of using ESCs, as they are not autologous and may induce immune reactions and rejection upon transplantation in the degenerated host retina.

#### 6.1.2. Preclinical Studies Using iPSCs

While ESC-based replacement therapy is valuable for retinal regeneration, it is complicated due to immune rejection, tumor formation, and ethical concerns. Therefore, several researchers investigated the use of induced pluripotent stem cells (iPSCs) for retinal cell-replacement therapy. Results demonstrated that iPSCs express various retinal progenitor cell-related proteins, such as *Pax6*, *Rx*, *Otx2*, *Lhx2*, and *Nestin* [52,53,54]. Direct differentiation of iPSCs into retinal ganglion (RG)-like cells was achieved with overexpression of *Math5* and the addition of DN and DAPT, with the cells surviving in the mice retina post-transplantation, but not integrating into the retina [54].

However, Venugopalan and colleagues [59] transplanted primary mouse RGC into uninjured mature rats’ retina in vivo by intravitreal injection and found results similar to using human iPSC and mesenchymal stem cells. The transplanted RGCs survived, migrated to the ganglion cell layer, and made functional synaptic connections in the host retina, responding to light stimulation [59]. The synaptic integration shows promise for allogeneic stem-cell-derived transplants as mice RGCs were successfully transplanted into rat retina [59]. Additionally, another study found that greater differentiation of iPSC-derived photoreceptors and purifying using fluorescence-activated cell sorting (FACS) allowed the cells to integrate into the outer nuclear layer and express photoreceptor markers after transplantation to the subretinal space of normal adult mice [55]—providing hope for autologous transplantation as a treatment for retinal degeneration.

Subretinal transplantation of iPSCs into retinal degenerative mice has resulted in iPSCs successfully integrating into the retinal outer nuclear layer and increased retinal function in hosts, as seen through electroretinographic analysis and functional anatomy [56]. The studies mentioned above have demonstrated the feasibility of photoreceptor replacement therapy using ESCs and iPSCs; however, transplant success based on disease stage remained unclear.

As mentioned before, a concern with cell transplants is immune rejection. To address this concern and oncogenic mutations, Sharma and colleagues [62] developed an oncogene mutation-free clinical-grade iPSC from AMD patients and differentiated them into RPE patches on biodegradable scaffolds. This allowed the cells to integrate into both rat and porcine models with AMD-like eye conditions [62]. On the other hand, another study of swine models mimicking end-stage AMD subretinal transplantation of hiPSC-derived RPE cells did not graft well in atrophic areas compared to healthy areas [67]. However, several engrafted RPE cells showed possible interaction with host photoreceptors as seen by the expression of immunolabeled phagosomes, suggesting a delay of the loss of visual function by decreasing GA progression [67]. A comparison of these two studies suggests that methods using scaffolds may provide more benefit and feasibility for autologous clinical-grade-induced RPE cell transplantation.

Human iPSCs-derived cells have also been shown to be effective in the long term post-transplantation. Human iPSC-retina grafts have been shown to survive up to 5 months in rats and up to 2 years in monkey models [57,63]. However, while some transplanted RGCs showed light responses in these models, it was not clear whether these responses were residual from the host retina or due to cell transplantation [63].

Another study reported that a combination of hiPSC-derived RPE cells and retinal precursor cells preserved endogenous photoreceptors and visual function, more than transplantation of either cell alone in early- and late-stage disease degeneration [65]. Further work is needed to investigate the benefit of a combination transplant and which cell combinations provide the most benefit.

#### 6.1.3. Preclinical Studies Using MSCs

One method to support autologous cell transplant and reduce immune rejection is through the use of mesenchymal stem cells (MSCs). Mesenchymal cells may be obtained from the bone marrow or adipose tissue of a particular patient and used as autologous cells for cell-replacement therapy. It has been shown that MSCs have anti-inflammatory properties, produce growth factors, and contribute to tissue regeneration, making them suitable for retinal degenerative cell therapy [60]. Additionally, MSCs can differentiate into RPE, photoreceptor-like, bipolar, and amacrine cells [43,48,69]. Recent studies have shown intravitreal injections of MSCs to have protective effects on the retina and enhance vision function [70,71]. On the other hand, a study using intravitreal or subretinal injections of bone marrow mononuclear stem cells reported increased cell survival, but no enhancement of retinal function in RCS and P23H-1 rats [72]. This calls for the need for further studies looking at various animal models. Another study observed photoreceptor regeneration and restoration of retinal function, following human adipose-derived MSCs in sodium iodate-induced retinal injury mice models, showing promise for MSC therapy in RP and AMD [73]. In addition to bone marrow and adipose-derived MSCs, they can also be derived from the umbilical cord. Notably, intravenously delivered small umbilical cord mesenchymal stem cells (average diameter 8.636 ± 2.256 µm) are safer and may protect visual function in RCS rats [69].

#### 6.1.4. Preclinical Studies Using Progenitor Cells

Preclinical studies using progenitor cells have shown them to restore some visual function in mice models. Klassen and colleagues [45] report successful engraftment of retinal progenitor cells in the degenerating retina of mature mice, with some cells maturing into neurons such as photoreceptors and expressing recoverin, rhodopsin, or cone opsin. Mice who received the transplant showed improved light-mediated behavior compared to controls [45]. Recently, He and colleagues [68] transplanted retinal progenitor cells from mouse ESC-derived retinal organoids and reported successful differentiation of transplanted cells, along with responses to light stimuli and integration with the host retina. While this study shows promising results, long-term effects and results need to be further investigated.

Moreover, while many studies before have used two methodologies to generate various retinal cells, one study created RPE and photoreceptor progenitor cells (PRP) cells using a single methodology. This unified protocol was created to generate RPE and PRP cells simultaneously from the same source of iPSCs, with cells surviving and integrating into rodent models of retinal degeneration post-transplant and improved vision [66]. This method provides an efficient and effective way to generate cells for combined transplantation. Further, intravitreal injection of human retinal progenitor cells (RPC) is shown to preserve retinal morphology but is only effective up to 12 weeks post-transplantation [64].

As there is a variety of stem cells that may be utilized in cell therapy for retinal degeneration, a study investigated which may be the most effective. It was reported that hiPSC-RPE cells have the best protective effect for retinal degeneration, transplanting better and longer than human adipose-derived stem cells, amniotic stem cells, bone marrow stem cells, dental pulp stem cells, and hiPSCs [74]. However, this finding must be further investigated, as there are differences between diseases, such as RP being genetic and AMD being related to older age, and studies with disease-specific animal models should be conducted. Preclinical studies have provided insight that stem cell therapy has the potential to stabilize or reverse progressive vision loss in both non-primates and primates, paving the way for clinical studies.

### 6.2. Clinical Trials

Most current clinical trials are currently in the early phases, focusing on initial responses and the safety of cell therapy (see Table 2). Current human trials are focusing on confirming that transplanted cells do not form teratomas, do not migrate into other organs, do not lead to immune rejection, and do not have other unintended adverse effects.

#### 6.2.1. Clinical Trials Using hESCs

Schwartz and colleagues conducted the first study investigating hESC-derived subretinal cell transplantation in human patients with Stargardt macular dystrophy and dry AMD (NCT01345006 and NCT01344993). Preliminary reports of two patients at 4 months post-transplant of hESC-derived RPE show that there were no signs of hyperproliferation, tumor formation, or transplant rejection [76]. A subsequent follow-up involving 18 patients, 9 with Stargardt and 9 with AMD, for a median 22-month period was reported. In this report, there were 10 eyes with an improvement in the Best Corrected Visual Acuity (BCVA) score, 7 eyes with a stable score, and 1 eye with a 10-letter decrease [77]. In addition, complications included cataracts in four eyes and the development of endophthalmitis in one patient [77]. These complications were reported to be attributed to pars plana vitrectomy surgery and the use of immunosuppressive treatment, not specifically with the hESC transplant [77].

While the participants in Schwartz and colleagues’ study were primarily white and black, W.K. Song and colleagues [78] investigated the safety and efficacy of hESC-derived RPE transplantation in four Asian patients: two with AMD and two with Stargardt disease (NCT01674829). Preliminary results were similar to Schwartz and colleagues [77], with some visual acuity improvement in three patients and stable acuity in one patient one year post-transplant [78]. There was no reported adverse proliferation, tumor formation, or serious safety issues [78]. While there were some adverse reactions following immunosuppression, these stopped after cessation of the immunosuppression [78]. This study provided greater promise for the use of hESC-derived RPE transplantation in patients of various ethnicities.

Mehat and colleagues [79] found subretinal transplantation of hESC-derived RPE cells in 12 Stargardt patients to be safe and to result in no inflammatory reaction or uncontrolled proliferation (NCT01469832). There was evidence of subretinal hyperpigmentation in all 12 patients, suggesting survival and engraftment of the transplanted hESC-derived RPE cells [79]. However, they did not report any significant improvement or decline in retinal function by electroretinography post-transplant in any patient and only borderline BCVA improvements in four patients [79]. This was hypothesized to be due to the advanced stage of the disease at the start of the study and the slow rate of progression in Stargardt, suggesting protection against further deterioration may only be seen in a longer follow-up period.

A recent study in Korea has investigated the long-term safety of hESC-derived RPE transplantation in three Asian patients with Stargardt disease (NCT01625559). Sung and colleagues [80] found no serious adverse events to be present during a 3-year follow-up period, with improvement of BCVA in one patient and stable BCVA in the other two patients. Favorable function and anatomical results were reported, compared to the natural progression of Stargardt disease. Further, Li and colleagues [81] reported no adverse reactions in a longitudinal 5-year study investigating hESC-derived RPE subretinal transplantation in seven Stargardt patients (NCT02749734). While these studies show promise for the long-term safety and efficacy of subretinal hESC-derived RPE transplantation, further multicenter studies with a larger number of patients are needed. A study that is currently in progress aims to follow 36 patients for up to 10 years after an hESC-derived RPE cell subretinal transplantation (NCT03167203). A study in China is also investigating the treatment of dry AMD using hESC-derived RPE (NCT03046407); however, the results of this study and its progress are currently unknown.

Further work by da Cruz and colleagues [82] (NCT01691261) aimed to determine the feasibility and safety of using subretinal transplantation using a biocompatible hESC-RPE monolayer on a synthetic basement membrane (a ‘patch’), rather than a suspension, in patients with wet AMD. Results from two patients show the stability of the hESC-RPE patch and improved BCVA and reading speed over 12 months [82]. As there were no control patients, the results of this study must be critically analyzed, but do suggest that an RPE patch transplant may be a beneficial form of treatment for retinal degeneration.

Preliminary safety results of a study (NCT03963154) investigating the use of a patch created using a novel tissue-engineered product consisting of hESC-derived RPE cells report successful integration in the retina, with no local inflammation or retinal deterioration observed in seven patients [83]. A recent study in Brazil is comparing whether surgical implantations of hESC-RPE monolayer on a polymeric scaffold or hESC-RPE injections into subretinal space are safer in AMD and Stargardt patients (NCT02903576).

Moreover, a five-year follow-up of a phase 1/2a clinical trial (NCT02590692) assessing scaffold-based hESC-derived RPE transplantation in 16 legally blind patients with GA reported the implant to be safe and tolerated [84]. The primary endpoint of the study was a safety assessment at 1-year post-transplant, which reported four patients in cohort 1 to have serious adverse events, including retinal hemorrhage, edema, retinal detachment, or RPE detachment [85]. However, these adverse events were mitigated in cohort 2 by using hemostasis during surgery. Patients were followed for a median of 3 years and reported a higher likelihood of BCVA improvements than worsening [84]. Patients who experienced worsening in BCVA had experienced the adverse events mentioned before during post-transplantation [84]. This study shows that scaffold-based transplants are successful and tolerated in patients with GA, suggesting this method as a possible treatment.

Recently, primary 24-month results from the currently active OpRegen hESC-derived RPE cell therapy trial (NCT02286089) suggest that subretinal transplantation of OpRegen is successful and safe. The data suggest that OpRegen counteracts RPE dysfunction and loss in GA [86]. Results report sustained BCVA gains at 24 months and greater improvement in retinal structure observed in patients with extensive coverage of GA with OpRegen and less advanced GA [86].

A study of unknown status in China is investigating the safety and efficacy of hESC-derived RPE cell subretinal transplantation in patients with RP (NCT03944239). An ongoing trial is also investigating the safety, tolerability, feasibility, and efficacy of retinal pigment epithelium stem cell (RPEESC)-derived RPE transplantation in patients with dry AMD (NCT04627428). The RPEESC is obtained from eyes donated to eye banks and the study aims to enroll 18 participants.

#### 6.2.2. Clinical Trials Using hiPSCs

While hESC-derived cell transplantations have proven to be safe and effective, there remains the concern of immune rejection. Hence, clinical trials are starting to be conducted utilizing hiPSC-derived cells, offering an autologous approach to cell transplantation.

Mandai and colleagues [87] reported iPSC-derived RPE sheets to be intact one-year after transplantation in one patient with AMD, but with no improvements in BCVA (UMIN000011929). Notably, the patient did not receive any immunosuppressants and there was no transplant rejection [87]. This same patient was then followed for a period of 4 years, showing survival of the RPE sheet and no adverse reactions [88]. This clinical trial showed promise for the use of autologous iPSC-derived transplantation for patients with retinal degeneration. However, there remains a need for larger study sizes and a variety of disease states to be investigated. The first of these trials in the United States is currently underway, investigating autologous transplantation of iPSC-derived RPE in AMD patients with GA (NCT04339764).

In India, investigators are evaluating the safety and efficacy of a novel hiPSC-derived formulation, Eyecyte-RPE, in patients with GA due to dry AMD (NCT06394232). This formulation is speculated to replace damaged RPE and potentially enable tissue regeneration.

A study in China is currently recruiting patients for autologous transplantation of hiPSC-derived RPE in AMD patients (NCT05445063).

#### 6.2.3. Clinical Trials Using MSCs

As mentioned before, MSCs provide paracrine effects and contribute to tissue regeneration, making them suitable for retinal degeneration cell therapy. Park and colleagues explored the safety and feasibility of intravitreal autologous CD34^+^ bone marrow cell injection in patients with AMD or RP (NCT01736059; NCT04925687). Preliminary findings of the pilot study conducted in six participants reported a single intravitreal injection to be well tolerated with no intraocular inflammation and no worsening of BCVA after 6 months [89]. These promising results led to a follow-up study conducted in RP patients to determine the number of CD34^+^ cells isolated for injection and adverse events (NCT04925687). Seven patients were enrolled in this study and a mean of 3.26 ± 0.66 million viable CD34^+^ cells were intravitreally injected in each eye [90]. While patients tolerated the injection 6 months post-injection, four patients had an extended follow-up and three of these four patients had progressive vision loss in both eyes [90]. Park and colleagues [90] note that it is unknown if repeat intravitreal injection of CD34^+^ would result in a greater therapeutic effect and larger studies are needed.

Another study investigated the safety of a single intravitreal injection of autologous bone-marrow-derived cells in patients with RP and the vision-related quality of life of these patients after the injection (NCT01068561; NCT01560715). Phase 1 (NCT01068561) results reported no adverse events associated with the injection over a period of 10 months [91]. Phase 2 (NCT01560715) results found there to be an initial improvement in the vision-related quality of life of these patients, but no difference from baseline at 12 months post-injection [92].

Tuekprakhon and colleagues [93] also investigated intravitreal autologous bone-marrow-derived MSC (BM-MSC) injection in 14 patients with advanced RP (NCT01531348). Their findings found improvements in BCVA initially, but BCVA returned to baseline at 12 months [93]. Researchers observed several patients with discomfort, such as mild pain, pressure, redness, and irritation, and mild adverse events, such as localized posterior synechiae, cystoid macular edema, and localized choroidal detachment, in their 12-month follow-up period [93]. One patient experienced a serious adverse event (diffuse vitreous hemorrhage) 3 years post-injection and required surgery, after which, vision was restored for the patient [93]. The adverse reactions and little improvement in visual function results warrant further investigation of BM-MSC injections in patients with RP.

On the other hand, Siqueira and colleagues [94] investigated intravitreal autologous bone-marrow-derived stem cell injection in patients with dry AMD (NCT01518127). Data reported intravitreal injections to be safe in patients with dry AMD and showed that there were slight increases in BCVA three months after injection [94]. We wonder if a longer-term follow-up period would result in findings similar to Tuekprakhon and colleagues [93], with BCVA returning to baseline.

The Stem Cell Ophthalmology Treatment Study (SCOTS and SCOTS2) is a multicenter trial investigating autologous BM-MSC treatment for the treatment of retinal disease and optic nerve damage (NCT03011541). This study aims to recruit 500 participants and follow them for a 12-month period. Weiss and Levy have reported findings of Stargardt disease, AMD, and RP patients. In the study, 34 eyes with Stargardt disease received autologous bone marrow injections using retrobulbar, sub-tenons, intravitreal or subretinal, and intravenous injection [95]. Over one year, statistically significant results (*p* = 0.0004) showed 21 (61.8%) eyes to improve, 8 (23.5%) to remain stable, and 5 (14.7%) to continue to have disease progression [95]. Visual acuity improvement was also seen in some patients [95]. Similarly, there were significant clinical improvements in visual acuity and a delay in vision loss seen in 32 patients with AMD [96]. In the 33 patients with RP, there were also improvements in visual acuity and stability of disease progression seen over a follow-up period of at least 6 months [97].

A phase 3 clinical trial investigated the management of RP using Wharton’s jelly-derived MSCS (WJ-MSC) (NCT04224207). It was found that sub-tenon transplantation of WJ-MSCs was effective and safe during a 1-year follow-up period, in both autosomal dominant and autosomal recessive inheritance of RP [98].

Another clinical trial is currently enrolling participants for a study investigating the safety and efficacy of intravenous and sub-tenon delivery of allogeneic adult umbilical cord-derived MSC cells for the treatment of RP (NCT05147701). Adverse effects will be monitored for a four-year follow-up period.

#### 6.2.4. Clinical Trials Using Progenitor Cells

We did not come across many clinical trials investigating the use of progenitor cells. A clinical trial is currently enrolling participants in a study investigating retinal stem and progenitor cell therapy for the treatment of AMD (NCT05187104).

## 7. Future Directions

There have been numerous advances in the utilization of cell therapy for retinal degenerative diseases. Progress continues to be made in this field with new advances in cell therapy techniques and combinations of therapies. However, there remain challenges in translating research findings to clinical populations and ethical sourcing of stem cells. In this section, we will highlight progress made and outline some challenges that need to be addressed.

### 7.1. Advances in Cell Therapy Techniques

The method of delivering cells into the ocular region is critical to ensure cell survival and transplantation success. Currently, there are three methods that are commonly used to deliver cells into the ocular region: subretinal, intravitreal, and suprachoroidal injections. While subretinal injections enable direct effects on cells and tissue in the subretinal space, there can be complications such as retinal detachment [99]. On the other hand, although an intravitreal injection can be quite invasive, the vitreous is an immune-privileged site and shows promise for being a site of stem cell delivery. Wang and colleagues [64] found intravitreal injections to be safe to inject human retinal progenitor cells in RCS rats with no teratoma formation following injection and a delay in retinal degeneration, showing promise for clinical models. Another method of cell delivery has been the suprachoroidal injection. This method is less invasive and has high bioavailability, as it targets the choroid, retinal pigment epithelium, and neuroretina [100].

### 7.2. Combination Therapies

Integration of gene therapy and biomaterials shows promising advances in cell transplantation and retinal degeneration treatment. Gene therapy, specifically CRISPR-Cas9, enhances hESC survival by reducing cell immunogenicity and eliminating the need for immunosuppression [101]. The integration with gene therapy has also shown the success of autologous transplantation in Stargardt patients. CRISPR-Cas9 was used to correct the ABCA4 variant in hiPSCs of these patients and autologous transplantation was performed without any adverse effects [102]. This research shows that gene therapy in combination with cell therapy allows for in vitro gene editing and differentiation of retinal cells for autologous transplantation treatment of retinal dystrophy [102].

Additionally, biomaterials and scaffolds have been combined with cell therapy to optimize its results. The use of scaffold technology plays two major roles in cell therapy. One role is providing a platform to deliver a layer of cells and the other role is the delivery of drugs, promoting cell survival and integration, and immunosuppression [103]. For example, sometimes the transplanted RPE does not adhere well to Bruch’s membrane, resulting in the need for a scaffold to help the transplanted RPE adhere and differentiate [103]. Ideal scaffolds are biocompatible, biodegradable, and injectable [104]. Historically, gelatin substrate was used during photoreceptor transplantation to maintain the photoreceptor layer, as it dissolves at room temperature and is not neurotoxic [105]. Today, scaffolds are typically made of biomaterials such as parylene C, polyethylene, terephthalate, or poly (lactic-co-glycolic acid), and deliver cells in a more structured way, allowing for a better understanding of cell survival and differentiation [103]. Aside from scaffolds, biomaterial such as bone marrow has its own advantages in cell therapy. Bone marrow stromal cells migrate to sites of injury and can differentiate into various cells, including retinal cells, and produce neurotrophic factors to help with cell survival [27]. Bone marrow stromal cells can be used in autologous transplantation, reducing concerns of immune rejection [27].

Moreover, 3D bioprinting technology is being utilized to study retinal degeneration, and 3D-bioprinted eye tissue has been created using patient stem cells [106]. This bioprinted tissue will allow scientists to better understand AMD and its progression to wet AMD and allow for modeling of the disease process in vitro. The future of 3D bioprinting and its possibility to be used for therapeutic development provides an exciting area to develop further.

### 7.3. Barrier to Clinical Translation

Although the eye and subretinal space provide a unique immunological environment with immune privilege, there remain barriers to clinical translation. One of these barriers is the risk and occurrence of immunogenicity. As some cell therapy methods utilize allogeneic stem cells and embryonic cells, there is the possibility of host-mediated immune responses and allogeneic graft rejection post-transplant [39]. After a cell transplant, there is an innate immune response that mediates tissue stress and inflammation; sometimes, with this response, natural killer cells become activated and play a role in allogeneic cell rejection [39]. Some have attempted to use systemic immunosuppression to overcome graft rejection; however, this leads to the issue of increased infection. Recent advances are utilizing gene editing, such as CRISPR-Cas9 to reduce immunogenicity, but there remains a need to ensure the safety of using gene-edited cells in clinical settings [101].

As mentioned before, a method to overcome the issue of graft rejection and promote cell survival and integration is to use iPSCs. Takahashi and colleagues [107] demonstrated the creation of induced pluripotent cells from adult human fibroblasts, providing evidence of the creation of patient-specific iPSCs. The use of iPSCs provides an autologous method for cell transplantation, decreases the risk of immune rejection, and eliminates the need for systemic immunosuppression [107]. However, the promise of iPSC in cell therapy is not without concerns as reprogramming of the cells raises concerns of genetic instability [108]. It has been stated that iPSC cells may have epigenetic memory and continue to proliferate, causing an increased risk of teratomas [39,109]. The unlimited differentiation possibility can also cause concern for the creation of human clones [110]. It is necessary that the use of stem cells for therapies be safety checked to ensure that there are benefits for the patients post-transplantation.

The clinical translation of cell therapy for retinal diseases faces several other challenges. Ensuring transplanted cells survive, integrate into the retinal tissue, and restore function is also critical, as is mitigating off-target effects that could lead to unintended complications [109]. Demonstrating meaningful functional recovery in vision and proving the long-term efficacy of these therapies are essential steps for clinical success. Scalability and standardization present additional barriers, as developing cost-effective, reproducible manufacturing processes that maintain consistent product quality is complex. Furthermore, cost remains a significant obstacle, with high therapy expenses limiting accessibility; ensuring broad patient access will require strategic efforts in cost reduction and healthcare reimbursement [29]. Finally, adequate and optimal delivery methods must be identified through clinical studies to ensure the correct and effective placement of cells within the retina [109].

### 7.4. Ethical Issues

The source of stem cells, in particular the derivation of pluripotent stem cells from human embryos and oocytes, has been a controversial topic amongst the clinical and public community since its utilization in medicine. However, iPSCs, which are reprogrammed from somatic cells, avoid ethical controversies as they do not use embryos or oocytes and are more commonly used today for cell therapy. While autologous stem cell transplantation minimizes ethical concerns and the risk of immune reaction, they raise concerns of genetic anomalies being present, a concern as retinal degenerative diseases have genetic origins [31]. As mentioned before, iPSCs do have complications of their own; therefore, it is important to optimize iPSC differentiation protocols to ensure that ethical concerns are mitigated.

## 8. Conclusions

This review has highlighted the promising potential of cell-based therapies in treating retinal degenerative diseases such as age-related macular degeneration (AMD) and retinitis pigmentosa (RP). We explored the anatomy and physiology of the retina, the pathophysiology of these diseases, and the various cell therapy approaches being developed, including embryonic stem cells (ESCs), induced pluripotent stem cells (iPSCs), mesenchymal stem cells (MSCs), and progenitor cells. Our focus on the most recent preclinical and clinical studies underscores the rapid advancements in this field.

Despite the progress, several barriers remain in translating these therapies into clinical practice. Challenges such as immune rejection, ensuring long-term survival and integration of transplanted cells, and the need for scalable and standardized manufacturing processes are critical areas that must be addressed.

For future research, we recommend a focused effort on overcoming these barriers to clinical translation. This includes developing strategies to enhance cell survival and integration, refining delivery methods, and addressing the immune responses that may arise from allogeneic transplants. Additionally, further exploration into the combination of gene therapy and biomaterials with cell therapy could provide a more comprehensive approach to treating retinal degenerative diseases. By addressing these challenges, the field of ocular cell therapy can advance toward developing clinically viable and widely accessible treatments, ultimately offering new therapeutic options for patients suffering from vision loss.

## Figures and Tables

**Figure 1 pharmaceutics-16-01299-f001:**
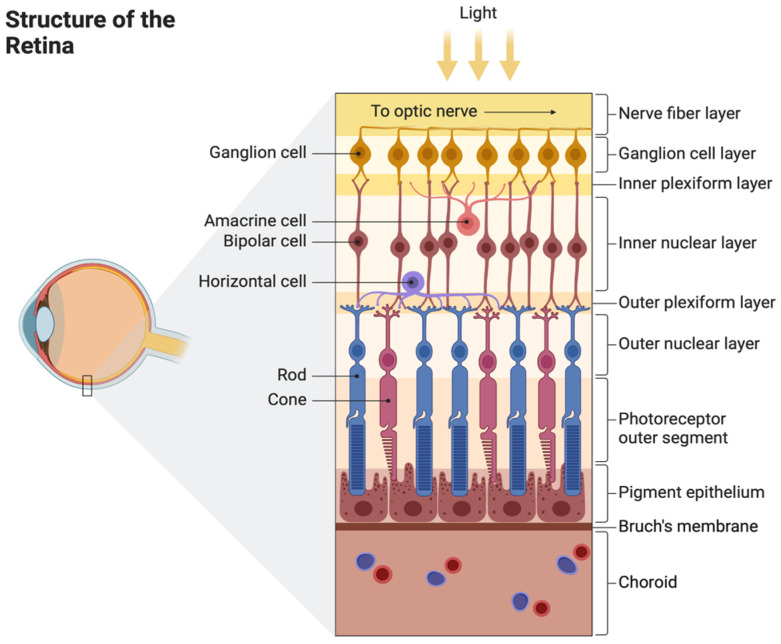
The retina consists of 10 layers: the inner limiting membrane, nerve fiber layer, ganglion cell layer, inner plexiform layer, inner nuclear layer, middle limiting membrane, outer plexiform layer, outer nuclear layer, external limiting layer, and photoreceptor layer. It also consists of 6 cell types: ganglion, amacrine, bipolar, horizontal, and photoreceptor cells. Reprinted from “Structure of the Retina”, by BioRender.com (2024). Retrieved from https://app.biorender.com/biorender-templates accessed on 15 August 2024.

**Figure 2 pharmaceutics-16-01299-f002:**
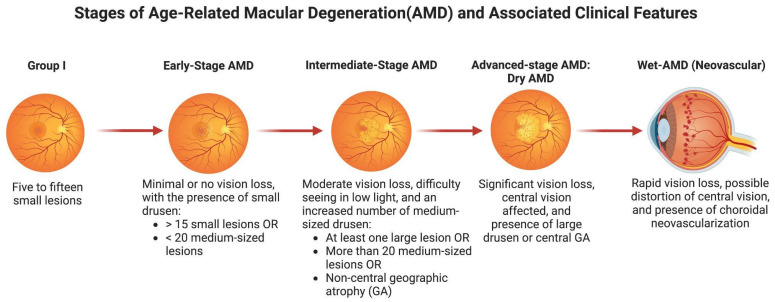
The progressive stages of age-related macular degeneration (AMD) and the associated clinical features. Adapted from the “Non-Alcoholic Fatty Liver Disease (NAFLD) Spectrum”, by BioRender.com (2024). Retrieved from https://app.biorender.com/biorender-templates, accessed on 15 August 2024.

**Figure 3 pharmaceutics-16-01299-f003:**
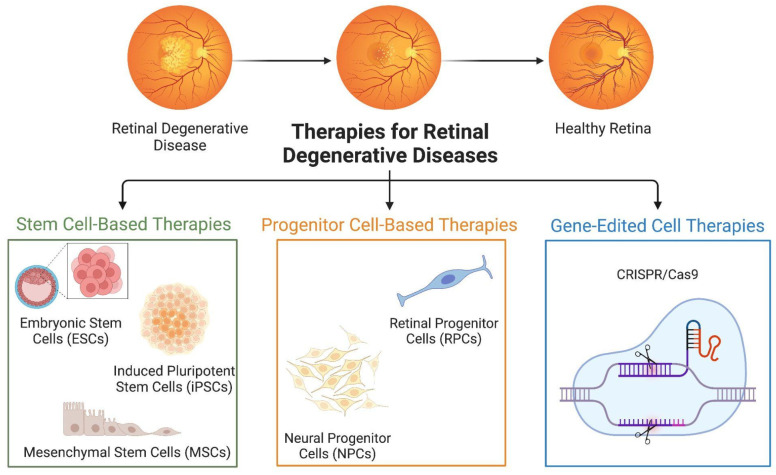
Cell-based therapies for retinal degenerative diseases. Adapted from “Immunotherapy Overview”, by BioRender.com (2024). Retrieved from https://app.biorender.com/biorender-templates accessed on 15 August 2024.

**Figure 4 pharmaceutics-16-01299-f004:**
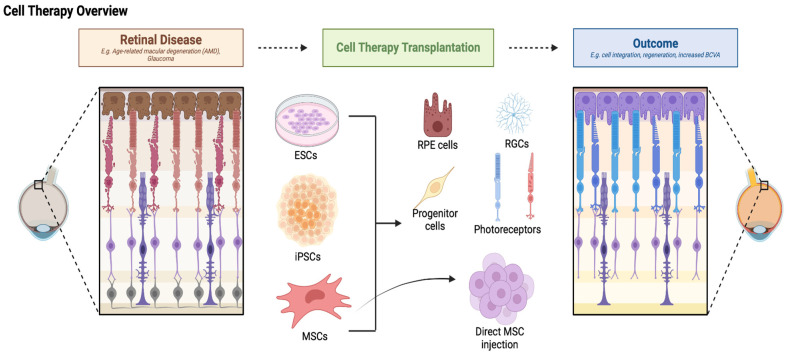
Preclinical and clinical trials to this date have utilized ESC, iPSC, MSC, and progenitor cell-derived cells to treat retinal diseases and improve functional outcomes. Adapted from “Retinal Disease and Regeneration”, by BioRender.com (2024). Retrieved from https://app.biorender.com/biorender-templates accessed on 15 August 2024.

**Table 1 pharmaceutics-16-01299-t001:** Preclinical trials investigating cell therapy treatment for retinal degeneration.

Disease Model	Animal Model	Protocol Description	Observed Effect	Reference
Retinal degeneration	Royal College of Surgeon (RCS) rats	Subretinal transplantation of donor RPE in host eye	RPE cells can be successfully transplanted into normal neonatal and adult rat eyes.	[40]
Retinal degeneration	Royal College of Surgeon (RCS) rats	Transplantation of donor RPE into subretinal space of dystrophic rat retina	Transplantation of RPE cells can prevent photoreceptor degeneration for at least 4 months.	[41]
Retinal degeneration	Royal College of Surgeon (RCS) rats	Subretinal transplantation of embryonic stem cells	Transplantation appeared to delay photoreceptor degeneration.	[42]
Retinal degeneration	Royal College of Surgeon (RCS) rats	Adult CD90 marrow stromal cells induced into cells with photoreceptor markers in vitro and then transplanted into RCS rats	MSC differentiated with autologous transplantation and integrated into the host retina with no teratoma formation.	[43]
AMD	Royal College of Surgeon (RCS) rats	Transplantation of RPE derived from primate ESC into subretinal space	Recovery of retinal function post-transplantation.	[44]
RP	C57BL/6 rho^−/−^ mice at 4 week of age or C3H rd mice at 4 weeks of age	Isolated retinal progenitor cells from day 1 eGFP transgenic CH7Bl/6 mice and expanded them; then, transplanted into mice with retinal degeneration	Donor cells integrated into retina and mice who received the transplant showed improved light-mediated behavior.	[45]
AMD	Royal College of Surgeon (RCS) rats	RPE derived from human ESC and transplanted into subretinal space of RCS rats	Cell survived in host, photoreceptors were restored, and vision improved.	[46]
AMD and RP	Royal College of Surgeon (RCS) rats	hESC-derived RPE was transplanted in the subretinal space of RCS rats	Cell survived in host, photoreceptors were restored, and vision improved.	[47]
Retinal injury and damage	10–12-week-old Wistar rats	Adult rat retinas underwent retinal damage via laser and then received bone marrow mesenchymal stem cell transplants	Bone marrow MSC survived in the retina and was incorporated into the outer nuclear layer, inner nuclear layer, and ganglion cell layer. Cells expressed rhodopsin and parvalbumin.	[48]
AMD and RP	Royal College of Surgeon (RCS) rats	RPE derived from human ESC and transplanted into subretinal space of RCS rats	Functional rescue in transplanted eyes compared to controls.	[49]
RP	Crx^−/−^ mice (model of Leber’s Congenital Amaurosis)	Retinal cells derived from human ESC were injected into mice retina using intraocular injection	Human ESC expressed markers for rod and cone photoreceptor cells once in subretinal space of mice and restored light response.	[50]
Glaucoma	ES cell culture from mouse D3-ES cells	Embryonic stem cells were differentiated in vitro and also transplanted in vivo	Embryonic cells can be used to treat degenerative diseases as they generate RGC-like cells in vitro and also differentiate into RGC cells in vivo after transplantation.	[51]
Retinal degeneration	hESC and iPSC	Provide a defined method of inducing hESC and iPSC into retinal progenitors, RPE, and photoreceptors	Induced retinal progenitor cells expressed RX, MITF, PAX6, and CHX10. Hexagonal pigmented cells expressed RPE65 and CRALBP. Photoreceptors expressed recoverin, rhodopsin, and phototransduction genes.	[52]
Retinal degeneration	hESC and iPSC	Determine whether hESC and iPSC model retinal development upon differentiation	Demonstrated that retinal cell specification from hESC and iPSC follows a sequence and time course similar to normal retinal development.	[53]
Glaucoma	BALB/c mice	Trying to see if induced pluripotent stem cells can express retinal progenitor cell genes and differentiate into retinal ganglion cells. Injected iPS-derived retinal ganglion-like cells into the retina	iPS cells express Pax6, Rx, Otx2, Lhx2, and Nestin genes inherently and over expression of Math5 and DN differentiate iPS into RG-like cells. Inhibiting Hes1 increases RGC genes. iPS-derived RG-like cells survive in retina but cannot integrate post-transplant.	[54]
N/a	Normal retina, adult wild-type mice	Generate iPSC with *OCT4*, *SOX2*, *NANOG*, and *LIN28* to derive photoreceptors for use in cell therapy for retinal transplantation	FACS-purified iPSC-derived photoreceptors can integrate into normal mouse retina and express photoreceptor markers.	[55]
RP	4–6-week-old dsRed-positive C57B1G mice were fibroblast donors and 4–6 weeks rhodopsin-null mice were transplant recipients	Adult dsRed mouse dermal fibroblast-derived iPSCs were transplanted in degenerative hosts	Cells formed teratomas. At 33 days, post-differentiation cells had markers for photoreceptors. CRX, recoverin, and rhodopsin. Increased retinol function in hosts with degenerative retina post-transplant.	[56]
RP	Monkey models	Determine ability of hESC-retina graft to transplant in rats and then conduct a pilot transplant in newly developed monkey models of retinal degeneration	Developed monkey models for study of retinal transplantation. Demonstrated hESC-retina graft to be effective in transplantation.	[57]
Retinal degeneration	Mouse models with mild degeneration (prom 1^−/−^) or severe degeneration (Cpfl1/Rho^−/−^)	Derived photoreceptors from organoids and subretinal transplantation in wild-type hosts	Retinal organoids had high photoreceptors and survived in the subretinal space of all mice. In mild degeneration cells integrated and had mature morphology. In the severe degeneration model, transplants remained in subretinal space and had rod-specific markers but no mature morphology.	[58]
Glaucoma	1–3-month Sprague Dawley rats	Transplanted GFP-labeled retinal ganglion cells into normal rat retinas by intravitreal injection	Cells integrated into the retina of adult rats (1–3 months) and made synapses post-transplantation.	[59]
Retinal degeneration	Female mice of inbred strain BALB/c age 7–9 weeks	Cultured MSCs to see growth factor expression, anti-inflammatory effects, and differentiation	Mesenchymal cells can differentiate into cells that show retinal markers, produce neuroprotective factors for retinal regeneration, and inhibit production of pro-inflammatory cytokines.	[60]
Retinal degeneration	4-week albino Royal College of Surgeon (RCS) rats	Isolated rat embryonic stem cells and induced them into retinal progenitor cells in vitro; transplanted into RCS rat retina	Visual function was restored in RCS rats. Potential clinical application of ESC cell therapy.	[61]
Retinal degeneration	Mice and pigs	Oncogene mutation-free iPSC was taken from AMD patients and differentiated into iPSC retinal pigment epithelium patches	Protocol was robust and efficient in generating RPE cells and rescuing degenerating retina in mice and pigs.	[62]
Retinal degeneration	Rhodopsin mutant SD-Foxn1 Tg (S334ter)3LacRrrc nude rats and 2 monkeys	Transplanted human iPSC retinas into animal models	Mature photoreceptors survived in the host retina for 5 months (rat) and 2 years (monkey). Some light responses detected in grafted areas in rats (4 of 7) and monkeys.	[63]
Retinal degeneration	BALB/c-mu mice	Transplanted human retinal progenitor cells via intravitreal injection into BALB/c-mu mice	Differentiated hRPCs had high retinal markers, no teratoma was formed, and retinal function improved. Slowed retinal degeneration. However, hRPCs were no longer effective 12 weeks post-transplant.	[64]
Retinal degeneration	Royal College of Surgeon (RCS) rats	Compared combined hiPSC-derived RPE and retinal precursor cell (RPCs) transplantation to either alone; in vivo monitoring conducted	Combined transplantation of hiPSC-derived RPE and RPC may be better than either transplant alone in retinal degeneration. Better visual response and conservation of outer nuclear layer.	[65]
RP	Royal College of Surgeon (RCS) rats	hiPSC-derived retinal cells and photoreceptor progenitor (PRP) cells transplanted in vivo via trans-scleral subretinal injection	Strong efficacy and safety for hiPSC-derived RPE and PRP cells in animals. No animal had teratoma formation and there was graft survival and integration. RPE transplant rescued vision function and there was functional photoreceptor activity.	[66]
AMD—geographic atrophy	Swine	Subretinal transplantation of hiPSC-derived RPE into healthy and degenerative retina areas	In vitro analysis showed the hiPSC-RPE cells to be differentiated, have typical epithelial morphology, and RPE-related gene expression. In the healthy retina, they engrafted and formed mature epithelium, but were patchy in atrophic areas.	[67]
Retinal degeneration	Royal College of Surgeon (RCS) rats	Transplanted retinal progenitor cells derived from mouse ESC-derived retinal organoids into RCS rats	The transplanted cells migrated to the inner retina and differentiated into photoreceptors, interneurons, and ganglion cells. The grafted cells elicited robust responses to light stimuli and integrated with the host retina.	[68]
RP	Royal College of Surgeon (RCS) rats	Derived umbilical cord mesenchymal stem cells (UCMSC) and then intravenously infused into RCS rats	Small UCMSC became stuck in lungs less and left quicker than UCMSC. Inflammation was inhibited and neurotrophic factors upregulated in retina and serum after transplantation. May be a potential therapeutic approach and delay degeneration in rats.	[69]
RP	*rd10* mice	Intravitreal injection of MSCs into mouse retina	Increase in survival rate of photoreceptors and visual function enhancement was observed through optomotor and electroretinogram responses.	[70]
RP	*rd12* mouse models with retinal degeneration	Intravitreal injection of adult MSC-derived RPCs into mouse retina	Transplanted RPCs led to improved vision and function. Observed anti-inflammation, retinal protection, and increased expression of genes involved in neurogenesis.	[71]
RP	Two animal models: RCS and P23H-1 rats	Utilized either intravitreal or subretinal injections of bone marrow mononuclear stem cell transplantations	Both forms of injections increased cell survival, as seen through mitigation of photoreceptor degeneration. No enhanced retinal function observed.	[72]
RP and AMD	Sodium iodate-induced retinal injury rat model	Transplantation of human adipose-derived MSCs	Transplantation facilitated photoreceptor regeneration and restoration of retinal function.	[73]
Retinal degeneration	3-week-old RCS rats	Compared subretinal transplant of stem cells, human adipose-derived stem cells, amniotic fluid stem cells, bone marrow stem cells, dental pulp stem cells, induced pluripotent stem cells, and hiPSC-derived RPE	Rats transplanted with any stem cell other than hiPSC had better visual function 4 weeks post-injection. Rats with hiPSC maintained visual function 8 weeks post-injection.	[74]

**Table 2 pharmaceutics-16-01299-t002:** Clinical trials investigating cell therapy treatment for retinal degeneration.

Trial Stage	Type of Cell Used	Disease	Sample Size	Approach	Country	Identifier
Phase I/II	hESC-derived RPE (MA09-hRPE)	SMD	13	Subretinal injection of 50,000–200,000 cells	USA	NCT01345006
Phase I and II—completed	hESC-derived RPE (MA09-hRPE)	Dry AMD	13	Subretinal injection of 50,000–150,000 cells in 5 cohorts	USA	NCT01344993
Terminated	hESC-derived RPE (MA09-hRPE)	Advanced Dry AMD	10	Transplantation of MA09-hRPE	Republic of Korea	NCT01674829
Phase I/II—completed	hESC-derived RPE (MA09-hRPE)	SMD	12	5 cohorts with 50,000–200,000 cell injections	UK	NCT01469832
Phase I—completed	hESC-derived RPE (MA09-hRPE)	Stargardt Macular Dystrophy	3	Subretinal transplantation of MA09-hRPE cells	Republic of Korea	NCT01625559
Phase I/II—unknown status	hESC-derived RPE	AMD and Stargardt	15	Subretinal transplantation	China	NCT02749734
Phase I/II—enrolling	hESC-RPE	AMD	36	Evaluating occurrence of late-onset adverse effects after hESC-RPE subretinal transplant	UK, USA	NCT03167203
Phase 1	hESC-derived RPE	RP	10	Transplant into subretinal space	China	NCT03944239
Phase 1—recruiting	PF-05206388—hESC-derived RPE	Wet AMD	10	Implantation of PF-05206388	UK	NCT01691261
Phase I and II—active	hESC-derived RPE	RP	12	Implantation of monolayer therapeutical patch into eye with worse acuity	France	NCT03963154
Phase I/II—completed	hESC-derived RPE	Dry AMD, wet AMD, and Stargardt disease	15	Compare the safety of surgical implantation of hESC-RPE monolayer on s polymeric scaffold versus hESC-RPE injections into subretinal space	Brazil	NCT02903576
Phase I/II—unknown status	hESC-derived RPE on parylene membrane (CPCB-RPE1)	Advanced dry AMD patients with geographic atrophy and central fovea involvement	16	Subretinal implantation of 100,000 differentiated RPE cells attached to a small parylene membrane	USA	NCT02590692
Phase I/IIa—active, not recruiting	OpRegen hESC-derived RPE	Dry AMD	24	Subretinal transplantation of 50,000–200,000 cells; see how cells engraft, survive, and moderate disease progression	Israel	NCT02286089
Phase I/II—enrolling	Retinal stem and progenitor cells	AMD	20	Cultured retinal stem and progenitor cells are injected subretinally	Belarus	NCT05187104
Phase I/II—unknown status	hESC-derived RPE	Dry AMD	10	Transplant into subretinal space	China	NCT03046407
Phase I/II—recruiting	RPESC-RPE-4W (allogeneic RPE stem-cell-derived RPE cells isolated from human cadaver)	Dry AMD	18	Patients will receive 50,000, 150,000, or 250,000 RPESC-RPE-4W cells in the macula of the eye.	USA	NCT04627428
Phase 1	Autologous iPSC-derived RPE	AMD	6	Determine safety of transplanting iPSC-derived RPE sheets	Japan	UMIN000011929
Phases I/IIa—recruiting	Autologous iPSC-derived RPE	Dry AMD	20	Subretinal transplantation of autologous iPSC-derived RPE in one eye	USA	NCT04339764
Phase I/IIa—recruiting	hiPSC-derived Eyecyte-RPE	Geographic atrophy secondary to dry AMD	54	Single-dose subretinal injection at varying doses: 100,000, 200,000, and 300,000	India	NCT06394232
Phase I—recruiting	Induced pluripotent stem cell (iPSC)	AMD	10	Autologous transplantation of iPSC-derived retinal pigment epithelium (RPE) into subretinal pace	Beijing	NCT05445063
Phase I	CD34^+^ stem cells from bone marrow	Irreversibly blind patients due to various retinal conditions	15	CD34^+^ bone marrow stem cells intravitreal	USA	NCT01736059 (pilot)
Phase I—completed	Autologous CD34^+^ stem cells harvested from bone marrow	RP	4	Intravitreal injection into 1 eye and followed for 6 months	USA	NCT04925687
Phase 1—completed	Autologous bone marrow stem cells	RP	5	Single intravitreal injection	Brazil	NCT01068561
Phase II—completed	Autologous bone marrow stem cells	RP	50	Single intravitreal injection	Brazil	NCT01560715
Phase I	Adult human bone-marrow-derived MSC	RP	14	Intravitreal injection	Thailand	NCT01531348
Phase I/II—completed	Autologous bone marrow stem cell	AMD or Stargardt with best-corrected ETDRS visual acuity <20/200	20	Intravitreal injection	Brazil	NCT01518127
Not noted	Autologous bone-marrow-derived stem cells	AMD, RP, Stargardt	500	Injection of autologous bone-marrow-derived stem cells	USA, United Arab Emirates	NCT03011541
Phase 3—completed	Umbilical cord Wahrton’s jelly-derived mesenchymal stem cells	RP	32	Cells implanted in sub-tenon space	Turkey	NCT04224207
Phase I—recruiting	Allogeneic adult umbilical cord-derived mesenchymal stem cells (UC-MSCs)	RP	20	Intravenous and sub-tenon delivery of 100 million UC-MSCs	Antigua and Barbuda, Argentia, Mexico	NCT05147701

## Data Availability

Not applicable.

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
