# Peer review of "Cell Therapy for Retinal Degenerative Diseases: Progress and Prospects"

_pharmaceutics, 2024, doi:10.3390/pharmaceutics16101299_

Round 1
Reviewer 1 Report
Comments and Suggestions for Authors
The review presented by .... is an excellent tool for scientists working in this field as a state-of-the-art point. The review is well done and follows a clear and well stated thread. I would advise the authors to be less redundant with the bibliography especially in the initial part of anatomical description of the retina and pathologies the same citations are given at the end of each sentence and in my opinion are unnecessary.
Author Response
Dear Reviewer,
We would like to express our sincere gratitude for your positive feedback and thoughtful suggestions on our manuscript. We truly appreciate your recognition of our work as an excellent tool for scientists in this field and your comments regarding the structure and clarity of the review.
In response to your valuable suggestion about reducing redundancy in the bibliography, we have carefully reviewed the manuscript and removed repeated citations, particularly in the initial anatomical description of the retina and related pathologies. We ensured that each citation now appears only once per paragraph, as recommended, to streamline the text while maintaining accuracy.
Thank you once again for your constructive feedback, which has helped us improve the quality of our manuscript. We hope the revised version meets your expectations.
Sincerely,
Reviewer 2 Report
Comments and Suggestions for Authors
Summary: This excellent review describes the recent progress made in retinal cell replacement strategies to reverse declines in retinal function resulting from retinitis pigmentosa (RP), wet and dry age-related macular degeneration (AMD), glaucoma, and Stargardt disease. It includes an overview of the anatomy and physiology of the retina, pathophysiology of AMD and RP. Subsequently, the unique advantages and challenges are described of different therapeutic outcomes of employing embryonic stem cells (ESCs), induced pluripotent stem cells (iPSCs), and mesenchymal stem cells (MSCs) in offsetting declines in the retinal phenotype resulting from RP and AMD. Mechanisms of action are described that underlie how different cell replacement techniques reverse declines in retinal function resulting from these diseases states. Furthermore, preclinical and clinical studies are described revealing the progress and obstacles that must still be overcome to translate these experimental procedures into usage in a clinical setting. Advances are described in cell delivery techniques, combination therapies, and integrating gene editing and biomaterials. This review will be an asset for investigators focused on designing novel approaches that improve treatment of retinal degenerative diseases.
Minor Concerns The review’s timeliness needs to be improved by including very recent publications that provide valuable data regarding the efficacy of various retinal cell replacements to reverse declines in retinal cell function in AMD, RP, glaucoma and Stargardt disease. In addition, there are several minor errors that need correction.
1) The stated focus of this study on line 45 is to describe the pathophysiology of AMD and RP and cell replacement therapy to reverse or stabilize disease their progression. However, they do not describe the pathophysiology of glaucoma and Stargardt disease even though they cite in Table 2 studies involving the outcome of cell replacement treatment of these diseases. It is necessary to indicate that the review also assesses cell replacement procedures to treat glaucoma and Stargardt disease. Furthermore, a description of their pathophysiology is warranted.
2) Line 178 deals with the limitations of anti vEGF therapy. Anti vEGF therapy improves visual acuity or at the least it stabilizes the neo vascularization response in a larger number of individuals. The authors need to provide a reference. [PMID 16154196 and 25598837].
3) Line 202 needs a reference citation. One possibility is PMID 38276971. Cell therapy for retinal disease. Goutham R Yalla and Ajay E Kuriyan.
4) Regarding Stargardt disease, RPE transplantation fails to directly address the etiology of the disease. PMI 39201545
5) Spelling correction on page 2 line 51: change discusses to discuss
6) Change oxygenation to oxygen on line 109 on page 3.
7) The outer blood retinal barrier modulates blood flow across the _______ while the inner retinal blood barrier controls flow across……
8) Muller should be upper case M rather than muller.
Comments on the Quality of English Language
English language usage is acceptable.
Author Response
Dear Reviewer,
We would like to thank you for your thorough review and positive feedback regarding our manuscript. We appreciate your kind words, as well as your constructive suggestions, which have helped us enhance the quality of our work. Below, we have addressed each of your comments:
- Recent publications: In response to your suggestion to improve the timeliness of the review, we have added more recent publications to the manuscript. These include references 70-73 in section 6.1.3 and reference 68 in section 6.1.4. After careful research, we did not find any additional recent clinical trials beyond those already included.
- Pathophysiology of glaucoma and Stargardt disease: We have now added sections 3.3 and 3.4, which cover the pathophysiology of both glaucoma and Stargardt disease. We have also updated line 45 to reflect that the review assesses cell replacement procedures for these two diseases.
- Line 178 reference: We have included the suggested references (PMID 16154196 and 25598837) to support the discussion of the limitations of anti-VEGF therapy.
- Line 202 reference: As requested, we have added the reference (PMID 38276971) to support the statement regarding cell therapy for retinal disease.
- Stargardt disease etiology: We have included the information regarding RPE transplantation and its limitations in addressing the etiology of Stargardt disease, citing PMID 39201545 in the newly added section 3.4.
- Spelling and grammar corrections: The spelling error on page 2, line 51 ('discusses' to 'discuss') and the correction from 'oxygenation' to 'oxygen' on line 109, page 3, have been made as suggested.
- Blood-retinal barrier clarification: We have reviewed the statement about the outer blood-retinal barrier and the choriocapillaris, and after consideration, we opted not to change it, as we believe it accurately reflects the anatomical structure described.
- Muller cells: We have corrected the capitalization of 'Muller' as suggested.
Once again, thank you for your detailed review. We hope that the revisions we have made address all your concerns and that the updated manuscript is now clearer and more comprehensive.
Sincerely,